# Validity and Reliability of the Leomo Motion-Tracking Device Based on Inertial Measurement Unit with an Optoelectronic Camera System for Cycling Pedaling Evaluation

**DOI:** 10.3390/ijerph19148375

**Published:** 2022-07-08

**Authors:** José Manuel Plaza-Bravo, Manuel Mateo-March, Roberto Sanchis-Sanchis, Pedro Pérez-Soriano, Mikel Zabala, Alberto Encarnación-Martínez

**Affiliations:** 1Department of Physical Education and Sports, University of Granada, 18071 Granada, Spain; jmplazabravo@gmail.com (J.M.P.-B.); mikelz@ugr.es (M.Z.); 2Sport Science Department, Universidad Miguel Hernández de Elche, 03202 Elche, Spain; 3Faculty of Sport Sciences, Universidad Europea de Madrid, 28670 Madrid, Spain; 4Research Group in Sports Biomechanics (GIBD), University of Valencia, 46010 Valencia, Spain; roberto.sanchis@uv.es (R.S.-S.); pedro.perez-soriano@uv.es (P.P.-S.); alberto.encarnacion@uv.es (A.E.-M.); 5Department of Physical Education and Sports, University of Valencia, 46010 Valencia, Spain; 6Department of Sports Sciences, Universidad Católica de Murcia UCAM, 30107 Murcia, Spain

**Keywords:** motion analysis, cycling, kinematics, range of movement, IMU

## Abstract

Background: The use of inertial measurement sensors (IMUs), in the search for a more ecological measure, is spreading among sports professionals with the aim of improving the sports performance of cyclists. The kinematic evaluation using the Leomo system (TYPE-R, Leomo, Boulder, CO, USA) has become popular. Purpose: The present study aimed to evaluate the reliability and validity of the Leomo system by measuring the angular kinematics of the lower extremities in the sagittal plane during pedaling at different intensities compared to a gold-standard motion capture camera system (OptiTrack, Natural Point, Inc., Corvallis, OR, USA). Methods: Twenty-four elite cyclists recruited from national and international cycling teams performed two 6-min cycles of cycling on a cycle ergometer at two different intensities (first ventilatory threshold (VT1) and second ventilatory threshold (VT2)) in random order, with a 5 min rest between intensity conditions. The reliability and validity of the Leomo system versus the motion capture system were evaluated. Results: Both systems showed high validity and were consistently excellent in foot angular range Q1 (FAR (Q1)) and foot angular range (FAR) (ICC-VT1 between 0.91 and 0.95 and ICC-VT2 between 0.88 and 0.97), while the variables leg angular range (LAR) and pelvic angle showed a modest validity (ICC-VT1 from 0.52 to 0.71 and ICC-VT2 between 0.61 and 0.67). Compared with Optitrack, Leomo overestimated all the variables, especially the LAR and pelvic angle values, in a range between 12 and 15°. Conclusions: Leomo is a reliable and valid tool for analyzing the ranges of motion of the cyclist’s lower limbs in the sagittal plane, especially for the variables FAR (Q1) and FAR. However, its systematic error for FAR and Pelvic Angle values must be considered in sports performance analysis.

## 1. Introduction

Biomechanics applied to cycling has undergone great evolution in recent decades, both in the analysis techniques and in the technological systems used. This evolution has allowed the passage of traditional techniques for different bicycle adjustments such as the inseam leg length measurement multiplied by a correction factor [1], the use of static goniometers to measure knee angle [2], or 2D and 3D motion capture systems as a gold standard [3]. All these techniques mentioned above start from an important limitation, isolating the cyclist from the ecological context of their sports practice and leading them to develop their capabilities under laboratory conditions.

Recently, the advancement in the technology of different wearables has allowed us to capture and analyze human movement in an ecological way with hardly any interference in it [4,5]. This is the case of the Inertial Measurement Units (IMUs) devices consisting of accelerometers, gyroscopes, and magnetometers, which allow us to measure accelerations and angular velocities of the body segments to which they are attached [6].

At present, this type of technology is widely used, and currently, this type of device is becoming more and more frequent in the analysis of different joint angles [7] applied to human gait [8], especially to analyze heel strike and toe off [9], in the running analysis [10,11], in weightlifting to measure barbell speed [12], in the learning and improvement of the swimming technique [13], and recently also in cycling [14]. All these examples show that technological advances have given researchers the ability to study movement in a more movement-friendly manner.

Knowing the relationship between the movements of the ankle, knee, and hip joints is presented as necessary as it could help us to better understand the mechanisms of power production and how the adjustment of these different variables can condition sports performance or comfort in cycling [15,16]. However, despite the widespread use that is made today of this type of device for measurement of the cycling kinematics of different joints in the analysis of human movement [10,12,17], we know that the precision of these devices depends on the task to be measured [18], as well as the anthropometric characteristics of each person [19] and the software that you use to filter the raw data collected by the sensors. That is why each IMU device must be validated individually. The extended use of this technology without prior validation may lead to the wrong interpretation of the results obtained. The incorrect interpretation could be associated with the loss of sports performance or injuries in the worst of cases, so the results of this study will indicate whether the use of this technology is valid and reliable in the cycling environment.

Among the current measurement systems through IMUs, the Leomo system (TYPE-R, Leomo, Boulder, CO, USA) is being used more and more by the cycling community, scientists, and experts in bicycle adjustment (bike fitting), as it is a system that allows the most ecological and timely assessment of real kinematics of the athlete. From a practical view, the Leomo system allows kinematic data collection in a real environment with no complex instrumentation required and is quick and easy to use for cyclists and coaches. For cyclists, that system does not interfere with pedaling, and does not disturb the cyclist with the sensor’s complex fixation that happens with markers employed in 2D or 3D cameras systems. This system would allow coaches to measure in a real situation, and affords practical kinematic information related to performance and cycling technique. It could also help in technique training based on the biofeedback process without the need of complex biomechanical training.

As a disadvantage, the system only evaluates a few variables, without allowing the inclusion of more variables of interest or the assessment of other anatomical locations of interest, such as the position of the arms and trunk. Likewise, it requires the use of a console to record the data. Similarly, the processes of calibration, filtering, and extraction of information are protected by the trademark, and the exact protocol is unknown.

Nevertheless, we think that the Leomo system may be able to measure correctly and provide values not known until now in field tests that can improve the science of cycling. Traditionally, the hip, knee, and ankle joints have been analyzed during the pedaling cycle; currently, within the sports performance field, Movement Performance Indicators (MPIs) have been included to explain performance and distinguish cyclists’ technique [20]. Despite their widespread use among professional cycling teams, MPIs assessed using IMU technology have not previously been validated against valid and reliable technologies such as gold-standard 2D or 3D camera systems.

We hypothesized that the Leomo system would be a valid and reliable system for accurately measuring the kinematics of the lower-limb joints in the sagittal plane during pedaling. Therefore, the objective of this study was to evaluate the reliability and validity of the Leomo system (TYPE-R, Leomo, Boulder, CO, USA) to measure the angular kinematics of the lower extremities in the sagittal plane during pedaling at different intensities compared to a 3D motion capture system (OptiTrack, Natural Point, Inc., Corvallis, OR, USA) considered as a gold standard. The key contributions of this paper can be summarized as follows:We checked the validity and reliability of a well-known and commonly used IMU in a cycling performance context in the search of a more ecological situation.We compared the agreement of the IMU with a gold-standard motion capture system.We described the bias of the IMU system, and we give several recommendations to mitigate the lack of agreement between systems to allow coaches to interpret the results in a performance context.

Next, in Section 2, the materials and methods used in this study are presented, while, in Section 3, the results are presented through tables and figures to facilitate their understanding. Next, in Section 4, the results of the present study are discussed with previous studies, and the manuscript ends with a section in which the main results are synthesized in the form of conclusions of the study.

## 2. Materials and Methods

### 2.1. Participants

The Human Research Ethics Committee of the University of Valencia approved the study (registry number: 20390). Twenty-four male elite cyclists were recruited from national and international cyclist club teams in December 2020. Inclusion criteria were: to be selected by the National Cyclist Federation, to be injuries-free in the last half year; not to be taking medication that alters normal cycling; not to suffer musculoskeletal disorders, neurological disorders, or heart failure that could affect normal pedaling. Cyclists were excluded if they have had significant injury, illness, or surgery within the previous six months. Before their inclusion in the study, all participants provided informed consent. A sample size calculation was performed based on the related sample comparison design, using the G-Power 3 software (version 3.1.9.7, Düsseldorf, Germany). This analysis indicated that at least a sample of 16 cyclists was required to detect significant differences in the different variables analyzed with a minimum detectable effect size of f = 1 (large) (α = 0.05, β = 0.05, and power = 0.96).

### 2.2. Experimental Designs

Before testing sessions, the anthropometric characteristics of the cyclists were taken by a bioelectrical impedance analysis (BIA) system, dual-frequency (50 kHz and 6.25 kHz) (Tanita BC-545, Tanita Europe BV, Amsterdam, The Netherlands). Cyclists were required to stand barefoot on metal electrodes while grasping two electrodes fixed on a handle with the arms extended. The percentage of body fat was calculated using all scales. Anthropometric measurements were performed by the same trained researcher who followed the device manufacturers’ recommendations. The anthropometric characteristics of the sample are shown in Table 1.

Lower-limb angular kinematic data during cycling were recorded using an optoelectronic three-dimensional (3D) camera system (Optitrack V120:Trio, NaturalPoint, Inc., Oregon, OR, USA), at a sampling frequency of 120 Hz, which was placed perpendicularly 7 m away from the sagittal plane of the cyclist. Simultaneously, a system consisting of a set of five inertial-measurement-unit (IMU) motion-tracking lightweight sensors (12 g) (Type-S Motion Sensor, Leomo, Inc., Boulder, Colorado, CO, USA), placed on the skin of cyclists and sampling at 100 Hz, was used.

Validity and reliability were evaluated at two common training intensities. The power, measured in watts (W), developed at aerobic ventilatory thresholds (first ventilatory threshold (VT1) and second ventilatory threshold (VT2)) were calculated by an incremental ramp test [20] 48 h before testing. Then, cyclists performed two bouts of 6 min pedaling at different intensities (VT1 and VT2) in a randomized order, with a 5 min rest between intensity conditions to avoid fatigue interference. In each intensity condition, two 30 s independent datasets were taken, separately by at least 2 min during the last four minutes at a controlled cadence of ±90 rpm and with their hands on top of the handlebars throughout the testing session (Figure 1). The first repetition intended to assess the validity of the IMU sensor system (Leomo) for angular kinematics assessment versus a gold-standard for angular kinematics assessment, an optoelectronic 3D camera system (Optitrack); the second repetition intended to test the reliability. Both cycling kinematics’ measurement repetitions were undertaken in the biomechanics lab under the same environmental conditions and at the same time of the day. Rate of Perceived Exertion (RPE) [21] was also registered after each cycling intensity condition (VT1 and VT2) and each repetition (REP 1 and 2).

A randomized design protocol was used to determine the intensity condition order. For this purpose, opaque envelopes were used for allocation concealment [22]. As many envelopes as possible orders of the study conditions were prepared with the sequence of study conditions (two possible order conditions). These envelopes were placed, unmarked and unidentified, on the table in the laboratory and the cyclists randomly selected an envelope, determining the sequence of study conditions.

All tests were carried out using the same cycle ergometer (Wattbike Pro, Wattbike LTD, Nottingham, UK). Before testing, the geometric measurements of cyclists’ own bicycles were taken with the use of an anthropometric tape [23] to fit the cycle ergometer with the same bicycle configuration.

Before the warm-up, cyclists were instrumented with six retroreflective markers to the Optitrack system and with the five Leomo sensors. To avoid marker loss or Leomo sensors loss due to the cyclist’s sweat during pedaling tests, a complete fixation protocol was followed [24] and the Leomo sensors were protected under the cyclist’s clothes.

Six 10 mm retroreflective markers, placed in the dominant lower limb on the lateral aspects of the (1) posterior superior iliac spine (PSIS), (2) the anterior superior iliac spine (ASIS), (3) the greater trochanter, (4) the femoral condyle, (5) the lateral malleolus, and (6) the 5th meta-tarsal head, were tracked during each measurement with the Optitrack system.

Via previous sensor synchronization via Bluetooth, Leomo sensors were fixed, following the brand recommendations, at the sacrum, right above the tail bone; right and left thigh, centered and 10 to 15 cm from the top of the patella; right and left foot, on the cycling shoes’ laces closest to the tip of each foot using a sensor clip (Figure 2).

Before the kinematic assessment, a familiarization with the testing condition and a self-selected speed warm-up (10 min) were carried out [25].

### 2.3. Data Analysis

The XYZ Cardan sequence of rotations was used to calculate angular kinematics, and the 3D reconstruction accuracy was calculated by the root-mean-square error (RMSE). Results showed a systematic error for X (mediolateral), Y (anteroposterior), and Z (vertical) axes of 0.005, 0.012, and 0.037 mm, respectively.

To process the data, Motive software (NaturalPoint, Inc., Oregon, OR, USA) was used. In each repetition, a minimum of 45 complete pedaling cycles were registered approximately during the 30 s of recording. A fourth-order low-pass Butterworth filter with a cut-off frequency of 6 Hz was used to filter marker data [23].

Angular kinematics parameters were calculated using a custom routine performed with the MatLab R2020b program (Mathworks Inc., Natick, MA, USA).

To obtain the pelvic angle, the leg angular range (LAR), and the foot angular range during the first 90° of the pedaling cycle (FAR(Q1)) and during the whole cycle (FAR), the angle convention shown in Figure 2 was used. Therefore, the absolute angle of the sacrum with respect to the horizontal represented the Pelvic Angle. LAR represented the range of movement of the absolute angle of the thigh, calculated with the greater trochanter marker and the femoral condyle marker, during the whole cycle (LAR). The FAR represented the range of movement of the absolute angle of the foot, calculated with the lateral malleolus marker and the 5th meta-tarsal head marker, during the first 90° of the cycle (FAR(Q1)) or during the whole cycle (FAR). To define the anatomical position, the angular positions of body segments in a standing calibration trial were analyzed.

### 2.4. Statistical Analysis

Bland–Altman plots were used to check the agreement between systems for each variable. Differences in each variable between Leomo and Optitrack were plotted against the mean results [26]. Differences and the association between systems were assessed by the mean of a *t*-test and Pearson’s correlation coefficient, respectively. A two-way, random-effects, single-measure intraclass correlation coefficients model was performed to check reliability. According to previous studies [27], the standard error of measurement (SEM) and minimum detectable change values (MDCs), in combination with the ICC values, were calculated to assess the simultaneous validity between the Optitrack and Leomo systems, in addition to the within-systems test–retest reliability and measurement error over the two testing repetitions for all outcome measures [28]. ICCs were interpreted as: excellent (0.75–1), modest (0.4–0.74), or poor (0–0.39) [29]. SPSS Statistics (SPSS Inc. Version 26.0, Chicago, IL, USA) was used to carry out the statistical analyses. The equations reported previously by Jacobson and Truax [30] were used to calculate the MDC, which is also known as the reliable change index score, which is expressed as the percentage test–retest change in the angular kinematics parameter required to find a significant difference at an alpha level of 0.05 based on the Repetition 1 mean value.

## 3. Results

### 3.1. Perceived Exertion

Perceived exertion showed no differences (*p* > 0.05) between repetitions (rep 1 vs. rep 2) for any of the intensities (Table 2). However, significant differences were found as a function of the cycling intensity (VT1 = 276.0 ± 22.2 W vs. VT2 = 350.4 ± 24.1 W), with the perception at VT2 being high (*p* < 0.01, ES (η^2^) = 0.672; Mean dif. = −3.48; 95% CI = −4.6/−2.3).

### 3.2. Bland–Altman Plots

All participants successfully completed the two intensities and the two repetition measurements. The Bland–Altman plots for the Foot Angular Range (Q1), Foot Angular Range, Leg Angular Range, and Pelvic Angle during the VT1 intensity condition are provided in Figure 3. During the VT1 intensity condition, there was a small relationship between the difference and the mean for all the variables, except for pelvic angle parameter, which showed a low to moderate relationship (R^2^ = 0.4199). Likewise, all variables were overestimated (*p* = 0.000) with the Leomo system by a mean of 2.1° ± 1.8° for FAR(Q1), 3.6° ± 1.3° for FAR, 14.8° ± 5.1° for LAR, and, finally, 13.5° ± 9.8° for Pelvic Angle.

The Bland–Altman plots during the VT2 intensity condition are provided in Figure 4. The behavior observed during the VT2 intensity condition was very similar to that observed in VT1, and all variables were overestimated (*p* = 0.000) with the Leomo system by a mean of 1.7° ± 2.1° for FAR (Q1), 3.7° ± 1.3° for FAR, 15.1° ± 4.5° for LAR, and, finally, 12.8° ± 10.1° for Pelvic Angle, respectively.

### 3.3. Validity and Reliability

The results for the FAR (Q1), FAR, LAR, and Pelvic Angle during VT1 and VT2 intensity conditions are provided in Table 3 and Table 4, respectively. The results demonstrated that all variables measured with the Leomo system showed a bias toward higher values compared with the Optitrack system at any of the intensities (VT1 and VT2).

At VT1 and VT2, both systems showed excellent test–retest reliability (Table 2 and Table 3), with only the Leg Angular Range values being measured with the Optitrack system (VT1: ICC = 0.88 and VT2: ICC = 0.80, respectively), reducing slightly the level but reaching an ICC value over 0.75, considered as an excellent value. Concurrent validity was shown to be consistently excellent across FAR (Q1) and FAR variables and repetitions for VT1 (ICC = 0.91–0.95) and for VT2 (ICC = 0.88–0.97). Modest validity (VT1: ICC = 0.52–0.71 and VT2: ICC = 0.61–0.67) was shown on the LAR and Pelvic Angle, reaching the highest mean differences between systems, ranging between 12.0 and 15.0° difference. The SEM for all variables ranged from 1.46 to 3.92% in the Optitrack system, from 0.57 to 2.10% in the Leomo system at VT1, from 1.63 to 4.49% in the Optitrack system, and from 1.15 to 2.56% in the Leomo system at VT2, showing a good reliability for both systems. FAR (Q1) and FAR for both systems showed the best results related to SEM values, ranging from 1.29 to 1.86% at VT1.

The MDC in all variables ranged from 1.44 to 5.3% for the Optitrack system and from 0.89 to 3.6% for the Leomo system at VT1, whereas, at VT2, it ranged from 1.54 to 6.2% for the Optitrack system and from 1.79 to 2.8% for the Leomo system. The MDCs were slightly higher for both systems in the Pelvic Angle variable (5.3% at Optitrack and 3.6% at Leomo), and only in the LAR for the Optitrack System (4.1%). At VT2, MDC values were slightly higher at Pelvic Angle and LAR in the Optitrack system (6.2 and 4.9%, respectively). With respect to the other variables (FAR (Q1) and FAR), the MDCs were lower and similar in between for both systems (lower than 2%) at VT1, and between 1.7 and 2.4% (at VT2 on average) at Optitrack and Leomo systems, respectively.

## 4. Discussion

Thanks to the advancement of technology, cycling biomechanics has evolved notably in recent years, enabling the development of portable technology such as IMU devices, which allow us to measure the movement of the cyclist in their daily sports practice environment. These new devices are promising and open a wide range of possibilities, and although there are already several studies that have demonstrated the strength of IMU technology to measure joint kinematics [8,31], incorporating prior information and assumptions is necessary before drawing clinical decisions [14], and the validity and reliability of each device must be individually contrasted.

For this purpose, we carried out this study using five IMUs placed in the shoes, thighs, and sacrum of twenty-four cyclists with the objective of evaluating the reliability, validity and sensitivity of the angle measurements reported by the LEOMO system in comparison with a 3D photogrammetry motion capture system considered as the gold standard.

Leomo is a novel device, based on IMUs technology, that aims to provide real-time information on the position of the cyclist’s body segments during pedaling, and the fact that, for the first time, this information can be obtained in real time and in an ecological context highlights the importance of knowing the validity and reliability of the Leomo device. This fact will open a very important window to the knowledge of the angular kinematics in relation to the variables related to the internal and external load (i.e., heart rate, power, and torque) that can affect the performance of cyclists. Our results show that Leomo is a valid, sensitive, and reliable device when compared to a gold standard (Optitrack) to measure the lower-limbs angular kinematics in the sagittal plane while pedaling. Despite this, it should be considered that Leomo overestimates all the variables, especially the values for LAR and Pelvic Angle, in a range between 12 and 15°, respectively, at both intensities analyzed. These discrepancies were also reported by Dahl et al. [32], although they indicated a 5° smaller difference for the sagittal plane by the IMU device used in their study. These differences could be slightly corrected if a previous calibration protocol is applied [33]. However, this is not possible with Leomo, so, knowing this overestimation, the coaches and biomechanics can “correct” it when interpreting the results in their field interventions.

Regarding validity, our results indicate that both systems have high validity and are consistently excellent across FAR (Q1) and FAR, with both variables having an ICC ranging between 0.91 and 0.95 for the VT1 intensity condition and between 0.88 and 0.97 for the VT2 intensity condition. Conversely, the LAR and Pelvic Angle variables showed a modest validity, with the ICC ranging from 0.52 to 0.71 for the VT1 intensity condition and between 0.61 and 0.67 for the VT2 intensity condition. Our results partially agree with those achieved by Cho et al. [33] who reported ICC values greater than 0.938 for the ankle, knee, and hip flexion extension variables during walking.

The SEM ranged from 1.46 to 3.92% at VT1 and from 1.63 to 4.49% at VT2 in the Optitrack system, and from 0.57 to 2.10% at VT1 and from 1.15 to 2.56% at VT2 in the Leomo system, showing a good reliability for both systems. FAR (Q1) and FAR for both systems showed the best results related to SEM values, ranging from 1.29 to 1.86% at VT1. Furthermore, the data show MDC from 1.44 to 6.2% for the Optitrack system and from 0.89 to 3.6% for the Leomo system. In the case of Leomo, this presents a very high reliability in the variables FAR (Q1) and FAR, with a very low MDC (lower than 2.4%), which indicates that they are the best variables to use in the pedaling analysis, which is of great interest considering that the ankle joint must optimize its stiffness and maximize the effective transmission of mechanical energy to the crank for optimal pedaling [34].

As the main limitations of our study, we must be aware that the reliability and validity of this device outside the sagittal plane has not been explored, and other studies should delve into it. In other types of exercise different from pedaling, they have demonstrated reliability and validity [32]; however, there are other works such as the review of Poitras et al. [31] in which they speak of a lower validity for the measurement of abduction/adduction movements with respect to flexion movements. Chia et al. [35], despite supporting their reliability and validity, suggest the need for continuous validation and improvement of IMU systems, as well as a methodological improvement to further reduce measurement errors. Therefore, future research needs to explore the validity and reliability of measurements collected by Leomo outside the sagittal plane. Another aspect that is crucial in the quality of data is related with synchronization of the sensors with the central unit. Improving the protocol not only reduces the delay between sensor information, but also reduces energy expenditure [36,37]. It has been demonstrated that all systems had a delay depending on the synchronization protocol followed by each system [37], but it has been demonstrated that several delays do not represent a problem if all the sensors have the same delay, and it is recommended that systems have a common time reference [37]. Regardless, Leomo sensor’s internal data protocols are confidential, but the authors believe that they follow a time-based synchronization protocol. Future studies should analyze the effect of the possible delay with the kinematic response obtained using the analyzed system.

## 5. Conclusions

In summary, according to our results, Leomo is a reliable and valid tool to analyze the ranges of motion of the cyclist’s lower limbs in the sagittal plane, especially for the variables FAR (Q1) and FAR. However, even though its error is systematic, it must be considered that for the LAR and pelvic angle values, Leomo overestimates them between 12° and 15°. Therefore, it is necessary to improve the algorithms that these devices use to extract the data, and the results of these variables should be interpreted with caution.

## Figures and Tables

**Figure 1 ijerph-19-08375-f001:**
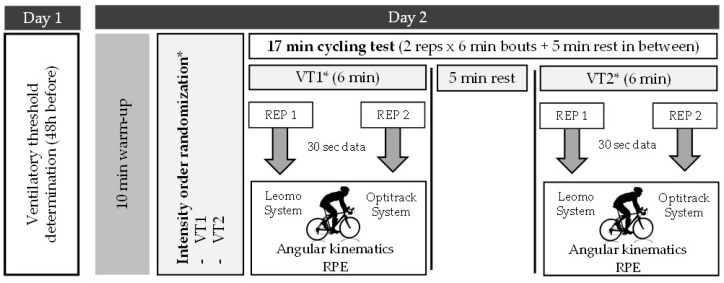
Study protocol procedure. * Intensity order was randomized.

**Figure 2 ijerph-19-08375-f002:**
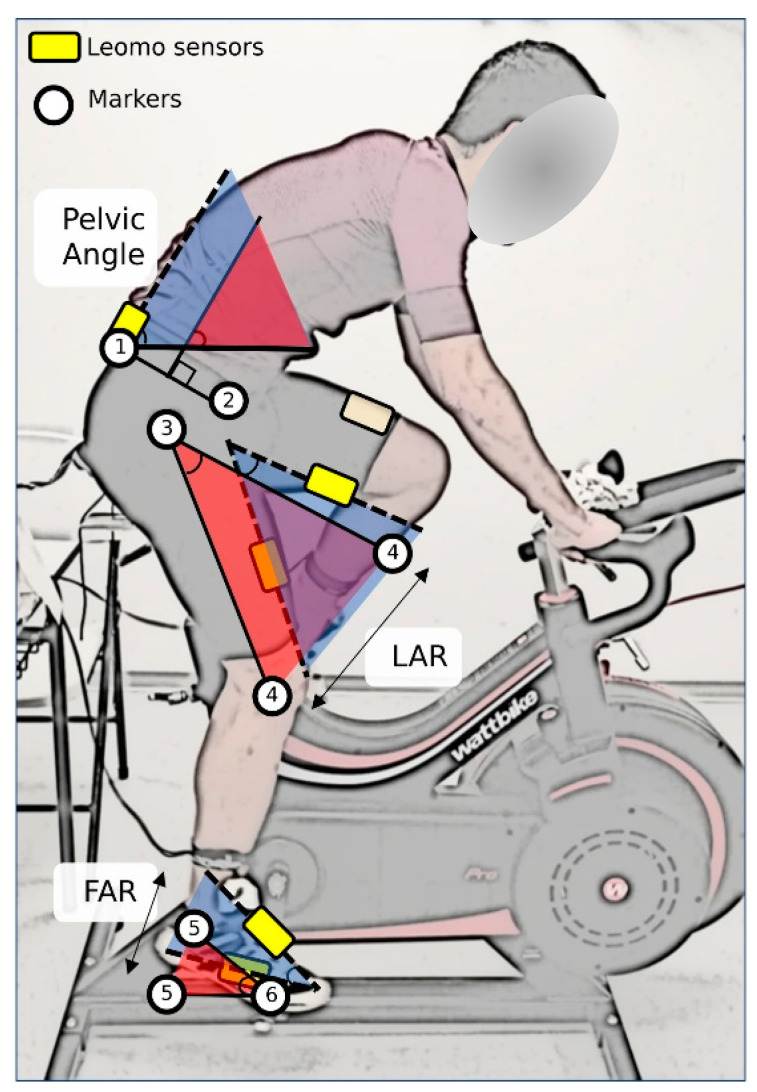
Conventions used for lower-extremity angles. White circle: retroreflective markers; yellow squares: Leomo sensors; 1: posterior superior iliac spine marker (PSIS); 2: anterior superior iliac spine marker (ASIS); 3: greater trochanter marker; 4: femoral condyle marker; 5: lateral malleolus marker; 6: 5th meta-tarsal head marker; LAR: Leg Angular Range; FAR: Foot Angular Range. Blue shadow represents the angular range measured by Leomo system; red shadow represents the angular range measured by Optitrack system.

**Figure 3 ijerph-19-08375-f003:**
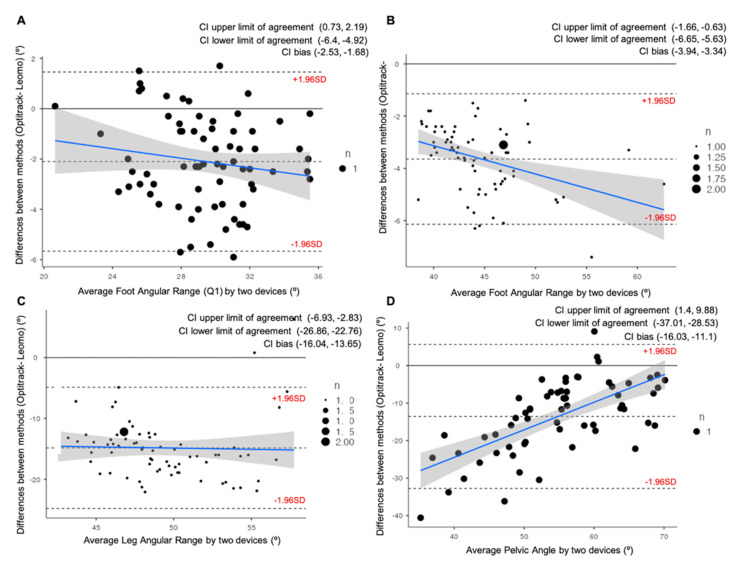
Bland–Altman plots representing comparisons between the Optitrack system and the Leomo system for four of the variables analyzed at VT1: (**A**) Average Foot Angular Range at first quartile of the pedaling cycle (Q1); (**B**) Average Foot Angular Range; (**C**) Average Leg Angular Range; (**D**) Average Pelvic Angle. The mean line represents the mean difference between the devices, with the upper and lower dashed lines representing the 95% limits of agreement (LOAs). 95% confidence intervals (CI) of the upper LOAs, lower LOAs, and bias are shown in the legend of each figure.

**Figure 4 ijerph-19-08375-f004:**
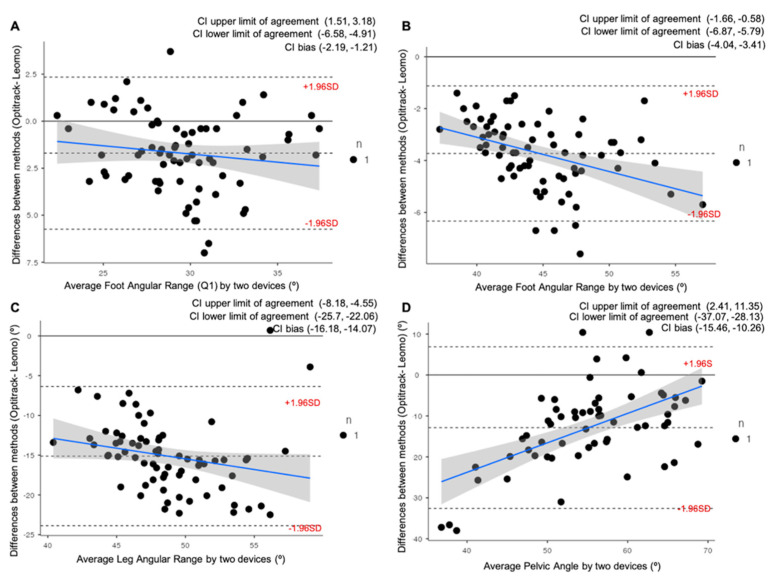
Bland–Altman plots representing comparisons between the Optitrack system and the Leomo system for four of the variables analyzed at VT2: (**A**) Average Foot Angular Range at first quartile of the pedaling cycle (Q1); (**B**) Average Foot Angular Range; (**C**) Average Leg Angular Range; (**D**) Average Pelvic Angle. The mean line represents the mean difference between the devices, with the upper and lower dashed lines representing the 95% limits of agreement (LOAs). 95% confidence intervals (CI) of the upper LOAs, lower LOAs, and bias are shown in the legend of each figure.

**Table 1 ijerph-19-08375-t001:** Anthropometric characteristics of the cyclists (means ± standard deviation).

Sample Profile (M ± SD)	
Age, years	20.0 ± 2.0
Weight, kg	68.1 ± 4.9
Height, cm	179.9 ± 6.0
BMI, kg/m^2^	21.1 ± 1.6
Fat mass (%)	9.3 ± 2.9
Water mass (%)	64.3 ± 2.7
FTP (W/kg)	5.14 ± 0.3

M = mean, SD = standard deviation, BMI: body mass index, W = watts, FTP: functional threshold power.

**Table 2 ijerph-19-08375-t002:** Rate of perceived exertion (RPE) descriptive values.

	Rep 1	Rep 2
VT1 (M ± SD)	11.8 ± 2.0	11.6 ± 2.1
VT2 (M ± SD)	15.1 ± 2.4	15.0 ± 2.4

M = mean, SD = standard deviation, VT1: ventilatory threshold 1, VT2: ventilatory threshold 2, Rep: repetition.

**Table 3 ijerph-19-08375-t003:** Validity and reliability of the Leomo system with Optitrack system for the assessment of angular parameters during pedaling cycle at VT1.

VT1	Optitrack	Leomo	Mean Diff (°) (95% CI)	ICC (95% CI)	Pearson
Foot AR (Q1) (°)					
Rep 1 (°) (M ± SD)	28.35 ± 3.00 *	30.46 ± 3.27	−2.11 (−2.53/−1.68)	0.91 (0.85/0.94)	0.836
Rep 2 (°) (M ± SD)	28.43 ± 3.15 *	30.36 ± 3.53	−1.93 (−2.37/−1.48)	0.91 (0.86/0.95)	0.844
*p* value (between reps)	0.521	0.471			
Mean Diff (°) (95% CI)	−0.08 (−0.33/0.17)	0.10 (−0.175/0.375)			
ICC (95% CI)	0.97 (0.95/0.98)	0.97 (0.95/0.98)			
SEM (% SEM)	0.52 (1.83)	0.57 (1.86)			
MDC (%)	1.44	1.57			
Foot AR (°)					
Rep 1 (°) (M ± SD)	43.6 ± 3.5 *	46.5 ± 4.8	−3.89 (−4.4/−3.4)	0.94 (0.90/0.96)	0.926
Rep 2 (°) (M ± SD)	42.6 ± 3.7 *	46.3 ± 4.8	−3.6 (−4.1/−3.2)	0.95 (0.92/0.97)	0.937
*p* value (between reps)	0.796	0.142			
Mean Diff (°) (95% CI)	−0.08 (−0.33/0.17)	0.10 (−0.17/0.37)			
ICC (95% CI)	0.97 (0.95/0.98)	0.98 (0.97/0.99)			
SEM (% SEM)	0.65 (1.46)	0.60 (1.29)			
MDC (%)	1.73	1.67			
Leg AR (°)					
Rep 1 (°) (M ± SD)	41.8 ± 4.4 *	56.7 ± 4.5	−14.84 (−16.0/−13.6)	0.52 (0.24/0.70)	0.353
Rep 2 (°) (M ± SD)	41.5 ± 4.5 *	56.5 ± 4.6	−15.01 (−16.1/−13.9)	0.61 (0.38/0.76)	0.443
*p* value (between reps)	0.406	0.165			
Mean Diff (°) (95% CI)	0.28 (−0.39/0.95)	0.11 (−0.05/0.27)			
ICC (95% CI)	0.88 (0.81/0.93)	0.995 (0.991/0.997)			
SEM (% SEM)	1.50 (3.58)	0.32 (0.57)			
MDC (%)	4.15	0.89			
Pelvic Angle (°)					
Rep 1 (°) (M ± SD)	48.55 ± 12.83 *	61.78 ± 6.47	−13.06 (−15.69/−10.44)	0.64 (0.41/0.78)	0.578
Rep 2 (°) (M ± SD)	48.50 ± 12.01 *	62.08 ± 7.02	−13.40 (−15.95/−10.85)	0.71 (0.50/0.83)	0.618
*p* value (between reps)	0.951	0.340			
Mean Diff (°) (95% CI)	−0.03 (−0.99/0.93)	−0.30 (−0.92/0.32)			
ICC (95% CI)	0.978 (0.962/0.987)	0.96 (0.93/0.97)			
SEM (% SEM)	1.90 (3.92)	1.29 (2.10)			
MDC (%)	5.27	3.59			

VT1: ventilatory threshold 1; M: mean; SD: standard deviation; AR: angular range; Rep: repetition; CI: confidence interval; ICC: intraclass correlation coefficient; Diff: difference; SEM: standard error of the measurement; MDC: minimum detectable change, expressed as a percentage of the Rep 1 mean value. * Differences (*p* < 0.01) between Optitrack vs. Leomo systems.

**Table 4 ijerph-19-08375-t004:** Validity and reliability of the Leomo system with Optitrack system for assessment of angular parameters during pedaling cycle at VT2.

VT2	Optitrack	Leomo	Mean Diff (°) (95%CI)	ICC (95%CI)	Pearson
Foot AR (Q1) (°)					
Rep 1 (°) (M ± SD)	28.64 ± 3.27 *	30.33 ± 3.54	−1.69 (−2.18/−1.20)	0.90 (0.84/0.94)	0.820
Rep 2 (°) (M ± SD)	28.80 ± 3.37 *	30.04 ± 3.84	−1.23 (−1.78/−0.69)	0.88 (0.82/0.93)	0.803
*p* value (between reps)	0.219	0.123			
Mean Diff (°) (95% CI)	−0.164 (−0.43/0.10)	0.293 (−0.081/0.667)			
ICC (95%CI)	0.97 (0.95/0.98)	0.95 (0.92/0.97)			
SEM (% SEM)	0.56 (1.95)	0.78 (2.56)			
MDC (%)	1.54	2.15			
Foot AR (°)					
Rep 1 (°) (M ± SD)	42.85 ± 3.84 *	46.73 ± 4.96	−3.87 (−4.34/−3.41)	0.95 (0.91/0.97)	0.931
Rep 2 (°) (M ± SD)	42.89 ± 4.18 *	46.42 ± 4.47	−3.53 (−3.87/−3.19)	0.97 (0.95/0.98)	0.945
*p* value (between reps)	0.847	0.170			
Mean Diff (°) (95% CI)	−0.03 (−0.37/0.31)	0.31 (−0.14/0.76)			
ICC (95% CI)	0.96 (0.95/0.98)	0.96 (0.93/0.97)			
SEM (% SEM)	0.69 (1.63)	10.0 (2.15)			
MDC (%)	1.94	2.78			
Leg AR (°)					
Rep 1 (°) (M ± SD)	41.16 ± 3.94 *	56.28 ± 4.82	−15.12 (−16.18/−14.07)	0.65 (0.45/0.78)	0.496
Rep 2 (°) (M ± SD)	41.37 ± 4.40 *	56.20 ± 5.06	−14.83 (−15.94/−13.72)	0.67 (0.48/0.80)	0.516
*p* value (between reps)	0.604	0.599			
Mean Diff (°) (95% CI)	−0.21 (−1.02/0.60)	0.08 (−0.23/0.39)			
ICC (95% CI)	0.80 (0.67/0.87)	0.98 (0.97/0.99)			
SEM (% SEM)	1.77 (4.29)	0.65 (1.15)			
MDC (%)	4.89	1.79			
Pelvic Angle (°)					
Rep 1 (°) (M ± SD)	49.76 ± 11.63 *	61.58 ± 6.68	−12.07 (−14.65/−9.47)	0.61 (0.35/0.77)	0.509
Rep 2 (°) (M ± SD)	49.29 ± 10.48 *	61.74 ± 6.79	−13.18 (−15.55/−10.81)	0.66 (0.42/0.80)	0.551
*p* value (between reps)	0.909	0.460			
Mean Diff (°) (95% CI)	−0.06 (−1.18/1.05)	−0.16 (−0.57/0.26)			
ICC (95% CI)	0.96 (0.94/0.98)	0.98 (0.97/0.99)			
SEM (% SEM)	2.24 (4.49)	0.89 (1.16)			
MDC (%)	6.2	2.48			

VT2: ventilatory threshold 2; M: mean; SD: standard deviation; AR: angular range; Rep: repetition; CI: confidence interval; ICC: intraclass correlation coefficient; Diff: difference; SEM: standard error of the measurement; MDC: minimum detectable change, expressed as a percentage of the Rep 1 mean value. * Differences (*p* < 0.01) between Optitrack vs. Leomo systems.

## Data Availability

The dataset generated and analyzed during the current study is available from the corresponding author on reasonable request.

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
