# Peer review of "Validity and Reliability of the Leomo Motion-Tracking Device Based on Inertial Measurement Unit with an Optoelectronic Camera System for Cycling Pedaling Evaluation"

_ijerph, 2022, doi:10.3390/ijerph19148375_

Round 1

Reviewer 1 Report

Dear authors,

It was a pleasure to read your manuscript in which validity and reliability of the Leomo motion tracking device has been investigated. this manuscript is well-written and informative. Results are clearly presented and then discussed. I have two suggestions which I think it would help readers to further benefit from reading this parer. 

1. What are the advantages/disadvantages of using existing and Leomo device? What are the practical barriers for using both systems?

2. Provide a more clear practical application for the target populations (i.e. coaches, athletes, etc..)

Author Response

[R1] = reviewer 1 comments / [A] = authors answers.

[A] Dear reviewer, thank you very much for your valuable comments. We hope the changes introduced in the manuscript satisfy your requirements and solves reviewer 1 doubts. We have included more information in the document to make the manuscript more understandable and clarify the reviewer 1 doubts. We work hard with two main objectives, the first one is to increase the knowledge to stimulate the safe sport practice, as well as to make accessible our small advances to all physical activity practitioners. So, any comments that help us to improve on our purpose is always welcome.

Responses to Reviewer 1

[R1] General comments:

[R1] It was a pleasure to read your manuscript in which validity and reliability of the Leomo motion tracking device has been investigated. this manuscript is well-written and informative. Results are clearly presented and then discussed. I have two suggestions which I think it would help readers to further benefit from reading this paper.

[A] Thank you for your comments. We have added that information in the introduction section, that try to answer the two following questions.

“From a practical view, Leomo system allow kinematic data collection in a real environment with no complex instrumentation required and being quick and easy to use for cyclists and coaches. For cyclists, that system don't interfere with pedalling, and don't disturb the cyclist with sensors complex fixation as happens with markers employed in 2D or 3D cameras systems. This system would allow coaches to measure in a real situation, and affords practical kinematic information related with performance and cycling technique. It also could help on technique training based on biofeedback process without needed of complex biomechanical training.

As a disadvantage, the system only evaluates a few variables, without allowing the inclusion of more variables of interest or the assessment of other anatomical locations of interest, such as the position of the arms, trunk, etc. Likewise, it requires the use of a console to record the data. Similarly, the processes of calibration, filtering and extraction of information are protected by the trademark and the exact protocol is unknown.”

[R1] 1. What are the advantages/disadvantages of using existing and Leomo device? What are the practical barriers for using both systems?

[A] Thank you for your comment, as reviewer 1 could read in the new information added to the introduction section, the advantages could be summarised as:

  • Kinematic data collection in a real environment
  • No complex instrumentation
  • Quick and easy instrumentation to use for cyclists and coaches
  • Don't interfere with pedalling
  • Don't disturb the cyclist with sensors complex fixation
  • Don’t need complex biomechanical training to use it

The disadvantages could be summarised as:

  • only evaluates a few variables
  • Don’t allow to assess other anatomical locations of interest
  • It requires the use of a console to record the data
  • The processes of calibration, filtering and extraction of information are protected by the trademark and the exact protocol is unknown

[R1] 2. Provide a more clear practical application for the target populations (i.e. coaches, athletes, etc..)

[A] Thank you for the suggestion. We have added that information in the introduction section. The practical application could be summarised as:

  • This system would allow coaches to measure in a real situation
  • Don't disturb the cyclist
  • Affords practical kinematic information related with performance and cycling technique
  • Help on technique training based on biofeedback

Reviewer 2 Report

Comments to the Author

This study aimed to evaluate the reliability and validity of the Leomo system by measuring the angular kinematics of the lower extremities in the sagittal plane during pedaling at different intensities. The research question is original and useful. This is an interesting paper with an adequate study design and methodology and the conclusions are supported by the results. Also, English language and style are fine. Then, I have some minor revisions to suggest: 

1. Introduction: 

- The introduction needs to be clear what the practical question is that you are trying to address. How the answer to this question is important to the field as this is not clear or obvious? How is this study and impactful study and not trivial as this needs more clarity as well. The key issue here is to make sure you set up your approach to the problem. The approach to the problem is essential in determining and describing the rationale for the study. You have not given a basic rationale for the choices made for the variables used in the study. Please treat to improve this part of the Introduction section.

2. Methods: 

- Have active females been researched? If not, why this study did not include females? If yes, should be introduced in the discussion session.

- Lines 108-110: Move the body composition procedures to 2.2. paragraph 

- For any equipment used to collect data that contributed to an analyzed variable, provide the model, manufacturer, and country, unless it is contained in the previously published methodology you have cited.

- There is a basic need to describe the technical characteristics of Tanita device. What is the calibration method to ensure validity (accuracy and precision) of the bioimpedance measurements? What is the technical error of measurement in vivo? Provide readers with a concise description of what this BIA device measures. In particular, what are the measurements detected by this tool? Do they directly measure the raw bioimpedance parameters (e.g., R, Xc, and phase angle)? Again, what equation was used to estimate Fat mass? Is it an equation developed using the Tanita device or an instrument that works with similar characteristics (frequency and technologies)? The equation should be reported.  

- It is important to consider that bioimpedance and bioimpedance-derived parameters (such as fat mass in this case) are dependent instrument and that the instrumental sensitivities are different. Therefore, no comparisons can be made between studies that measure phase angle with different technologies (e.g., foot-to-hand- or direct segmental in standing position) or sampling frequencies.

- Allocation procedures should be described. 

Results

- Table 1 should be revised removing extra-lines

Discussion:

- Again, it is not sufficiently justified how the field could be possibly strengthened by this study. The innovation of the study is not very clear to me.

Overall, the study may have the potential to add meaningful information to the current body of literature.

Author Response

[R2] = reviewer 2 comments / [A] = authors answers.

[A] Dear reviewer, thank you very much for your valuable comments. We hope the changes introduced in the manuscript satisfy your requirements and solves reviewer 2 doubts. We have included more information in the document to make the manuscript more understandable and clarify the reviewer 2 doubts. We work hard with two main objectives, the first one is to increase the knowledge to stimulate the safe sport practice, as well as to make accessible our small advances to all physical activity practitioners. So, any comments that help us to improve on our purpose is always welcome.

Responses to Reviewer 1

[R2] Comments to the Author

[R2] This study aimed to evaluate the reliability and validity of the Leomo system by measuring the angular kinematics of the lower extremities in the sagittal plane during pedaling at different intensities. The research question is original and useful. This is an interesting paper with an adequate study design and methodology and the conclusions are supported by the results. Also, English language and style are fine. Then, I have some minor revisions to suggest:

[A] Thank you for your comment.

[R2] 1. Introduction:

[R2] - The introduction needs to be clear what the practical question is that you are trying to address. How the answer to this question is important to the field as this is not clear or obvious?

[A] Thank you for your comment. We have added more information in the introduction section. The extended use of this technology without prior validation may lead to the wrong interpretation of the results obtained. The incorrect interpretation could be associated with loss of sports performance or injuries in the worst of cases, so the results of this study will indicate whether the use of this technology is valid and reliable in the cycling environment.

[R2] -How is this study and impactful study and not trivial as this needs more clarity as well. The key issue here is to make sure you set up your approach to the problem. The approach to the problem is essential in determining and describing the rationale for the study. You have not given a basic rationale for the choices made for the variables used in the study. Please treat to improve this part of the Introduction section.

[A] Thank you for the valuable comments. We have added more information in the introduction section trying to answer reviewer 2 requirements.

[R2] 2. Methods:

[R2] - Have active females been researched? If not, why this study did not include females? If yes, should be introduced in the discussion session.

[A] We have included the term "male" in the Participants section to clarify that the sample was made up entirely of men. Women were not measured in this study because access to the elite sample was complicated by the start of the season. The women of the National Cycling Team had already been evaluated and we could not have access to them. We believe that their inclusion in the studies is very important, but as it is a validation study, we consider that both the level and the number of participants were sufficient to achieve the objective of this study. However, we appreciate your suggestion and in future studies we will try to analyse female cyclists.

[R2] - Lines 108-110: Move the body composition procedures to 2.2. paragraph

[A] Thank you for your suggestion, that information has been moved to the 2.2. paragraph.

[R2] - For any equipment used to collect data that contributed to an analyzed variable, provide the model, manufacturer, and country, unless it is contained in the previously published methodology you have cited.

[A] Thank you for your suggestion. We have reviewed the document and have added in the introduction section the Leomo sytems information. After that citation, the next citation was as Leomo system to allow a better understanding and avoid repetition.

[R2] - There is a basic need to describe the technical characteristics of Tanita device. What is the calibration method to ensure validity (accuracy and precision) of the bioimpedance measurements? What is the technical error of measurement in vivo? Provide readers with a concise description of what this BIA device measures. In particular, what are the measurements detected by this tool? Do they directly measure the raw bioimpedance parameters (e.g., R, Xc, and phase angle)? Again, what equation was used to estimate Fat mass? Is it an equation developed using the Tanita device or an instrument that works with similar characteristics (frequency and technologies)? The equation should be reported. 

It is important to consider that bioimpedance and bioimpedance-derived parameters (such as fat mass in this case) are dependent instrument and that the instrumental sensitivities are different. Therefore, no comparisons can be made between studies that measure phase angle with different technologies (e.g., foot-to-hand- or direct segmental in standing position) or sampling frequencies.

[A] Thank you for your valuable comments. We have detected that the Tanita model was wrong, and we have change in the text. We also have added more information about BIA measurement and protocol. According with previous studies (https://doi.org/10.1155/2015/614858), our system showed a poor level of agreement (Lin’s concordance <0.9; 0.855) compared with body fat percentage measured by dual-energy X-ray absorptiometry. The system underestimates %BF, with a mean bias of −1.67%, ranging the LOA around -11% to 7%.

In that sense, our main objective in the study was to validate an IMU system with a gold standard camera system in measuring cycling kinematics. The anthropometric measurements were done to describe the sample population. Besides the bias of the system, which underestimates the %BF compared with DXA systems, our sample profile regarding %BF are in line with previous studies (10.1039/D0FO03456H).

[R2] - Allocation procedures should be described.

[A] Thank you for the suggestion. We have added the allocation procedures in the text.

“As many envelopes as possible orders of the study conditions were prepared with the sequence of study conditions. These envelopes were placed, unmarked and unidentified, on the table in the laboratory and the cyclists randomly selected an envelope, determining the sequence of study conditions.”

[R2] Results

[R2] - Table 1 should be revised removing extra-lines

[A] Thank you for the suggestion. We apologised for the mistake and have removed the extra-line.

[R2] Discussion:

[R2] - Again, it is not sufficiently justified how the field could be possibly strengthened by this study. The innovation of the study is not very clear to me.

[A] Thank you for your comment. We hope that with changes introduced in the introductions section, we have clarified the innovation of the study. Leomo system is being used among a wide range of cyclist population (elite and recreational) to improve performance and to prevent injuries, but no previous studies has been done to validate the Movement Performance Indicators (MPI) calculated by that system. With our results, we encourage that not all MPI variables are valid and reliable, as LAR and pelvic angle has a systematic error that need to be considered in practical applications, as they are overestimated around 12º to 15º. Whereas FAR (Q1) and FAR were shown to be valid and reliable.

[R2] Overall, the study may have the potential to add meaningful information to the current body of literature.

[A] Thank you for your comment. Our team works with that purpose, any comment that help us to improve our studies always will be welcome.
